



# Eddy Kinetic Energy Variability From 30 Years of Altimetry in the Mediterranean Sea

Paul Hargous[1], Vincent Combes[1,2], Bàrbara Barceló-Llull[1], and Ananda Pascual[1]

[1]Institut Mediterrani d'Estudis Avançats, IMEDEA (CSIC-UIB), Esporles, Spain
[2]Departament de Física, Universitat de les Illes Balears, Palma de Mallorca, Spain

**Correspondence:** Paul Hargous (hargous@imedea.uib-csic.es)

**Abstract.** Mesoscale activity plays a central role in ocean variability, substantially influencing the mixing of biogeophysical tracers, such as heat and carbon, and driving changes in ecosystems. Eddy Kinetic Energy (EKE), a metric used for studying the intensity of mesoscale processes, has recently been shown to increase in regions of intense EKE worldwide. Strong EKE positive trends are observed, for example, in the principal western boundary current regions, such as the Gulf Stream, Kuroshio Extension, and the Brazil/Malvinas Confluence. In this study, we assess whether the Mediterranean Sea, known to be a hotspot for climate change impacts, also exhibits such intensification. Despite the high number of observational data and modeling experiments, there is a gap in understanding the long-term evolution of mesoscale dynamics and EKE trends in the Mediterranean Sea. This study investigates EKE trends in the Mediterranean Sea using daily geostrophic currents derived from satellite altimetric data. To test the robustness of the results, we compare EKE trends computed from three different gridded altimetric products: a global product derived from a stable two-satellite constellation (two-sat) and two other products (global and European) incorporating all available satellites (all-sat). While all products reveal a general increase in EKE in the Mediterranean Sea over the last three decades, the trends calculated from the two-sat product are significantly smaller than those computed from the all-sat products. We show that this discrepancy is strongly linked to the increasing number of satellites over time used to construct the all-sat data sets, which enhances both spatial and temporal coverage, and hence, their capacity to detect higher energy levels. To evaluate the fidelity of these gridded products in capturing EKE trends, we compare them with along-track data in high-energy regions of the Mediterranean Sea: the Alboran Sea and the Ierapetra area. These regions exhibit contrasting EKE trends: positive in the Alboran Sea and negative in the Ierapetra area. These findings highlight the importance of using altimetric products with a stable number of satellites constructed for climate applications when addressing long-term ocean variability analysis.

## 1 Introduction

The ocean is a central component of the Earth's climate system, playing a key role in distributing heat and absorbing carbon. Since the beginning of the industrial era, the ocean is estimated to have absorbed more than $90\%$ of the excess heat generated by human activity and nearly one-third of anthropogenic carbon emissions (Abram et al., 2019). This massive uptake directly contributes to rising ocean temperatures, with consequences for sea level, thermohaline circulation, and overall climate dynam-



ics. In a climate change context, a deeper understanding of the response of the ocean is necessary.

A large part of the transport of water properties such as heat and nutrients in the ocean occurs at spatial scales ranging from 10 to 100 km, commonly referred to as the mesoscale (Chelton et al., 2011; Gaube et al., 2019; Becker et al., 2025). Mesoscale features (such as eddies, fronts, and meanders) are ubiquitous in the ocean and possess distinct physical and dynamical characteristics that enable them to efficiently transport these properties. A widely used metric to characterize mesoscale activity

and evaluate its intensity and variability is the Eddy Kinetic Energy (EKE), which quantifies the energy associated with time-varying flow. Recent studies have reported a robust global intensification of EKE, particularly in high-energy regions such as the western boundary currents, based on satellite observations (Martínez-Moreno et al., 2021; Barceló-Llull et al., 2025) and model simulations (Hu et al., 2020). These studies consistently identify positive trends in EKE within western boundary currents since 1993, year marking the beginning of the altimetry era. To evaluate the robustness of satellite-based EKE trends,

Barceló-Llull et al. (2025) compared two different satellite altimetry products and highlighted substantial differences in the derived trends. Barceló-Llull et al. (2025) compared a satellite product that merges all available satellite missions (all-sat) with another product based on a consistent two-satellite constellation and built for climate applications (two-sat), revealing that EKE trends derived from the two-sat product are significantly smaller than those obtained from the all-sat product.

Here, we focus on the Mediterranean Sea, a hotspot for climate change and an ideal miniature ocean for studying climate

change impacts (Bethoux et al., 1999; Escudier et al., 2021). It is a semi-enclosed basin characterized by relatively low EKE levels compared to the global ocean (Pujol and Larnicol, 2005), yet still rich in mesoscale eddies, filaments, and fronts (Barral et al., 2021; Mason et al., 2023; Zodiatis et al., 2023). The regional circulation is shaped by the inflow at the surface of Atlantic waters through the Strait of Gibraltar to compensate for the high evaporation (Schroeder et al., 2012), forming the Atlantic Jet (Renault et al., 2012). These Atlantic waters progress counterclockwise in the Mediterranean (Fig. 1), mixing with resident

Mediterranean waters and freshwater. The water input from the Atlantic becomes saltier, warmer and denser, during its journey before exiting through the Strait of Gibraltar, in depth. A more detailed description of surface currents is provided by Escudier et al. (2021). The Atlantic Jet drives anticyclonic gyres in the Alboran Sea (Renault et al., 2012) (Fig. 1). These features, along with other recurrent structures such as the Ierapetra eddy (Fig. 1) in the eastern basin (Ioannou et al., 2017), plays a central role in the Mediterranean's mesoscale dynamics. A key question is whether these energetic features have intensified

over the altimetry period, in line with the increasing EKE observed in several regions worldwide (Martínez-Moreno et al., 2021; Barceló-Llull et al., 2025).

The objective of this study is to investigate the spatial distribution and temporal evolution of EKE across the Mediterranean basin. To assess the robustness of the EKE variability, we compare three gridded altimetric products which differ in their satellite constellation configurations and resolution, highlighting the role of observational coverage in detecting mesoscale activity.

We evaluate the fidelity of these products in capturing EKE trends by comparing them with along-track data (L3). Our results reveal spatial heterogeneity in EKE trends, with contrasting behaviors in energetic regions such as the Alboran Sea and the Ierapetra area.



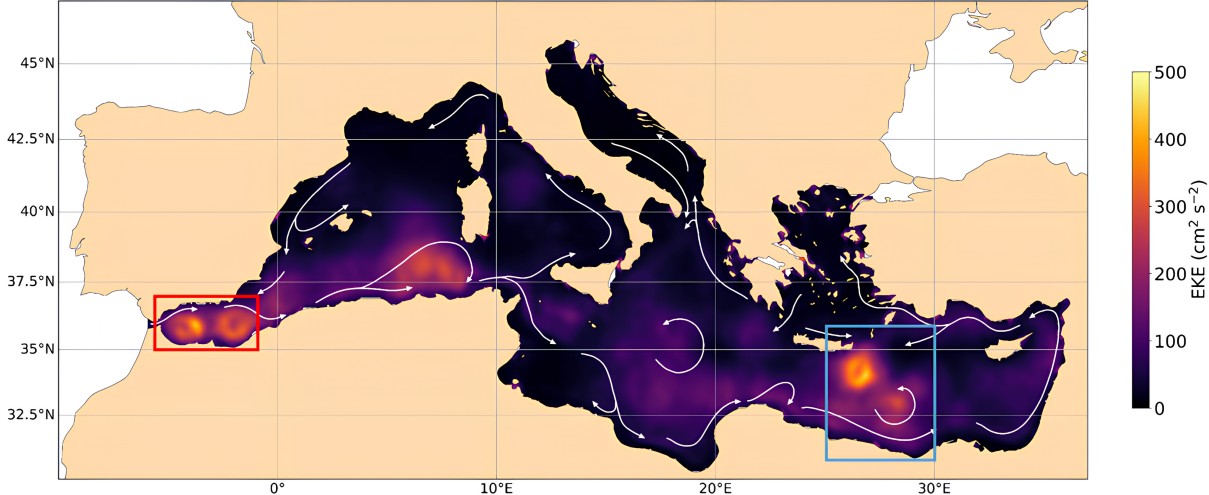

**Figure 1.** Temporal average of EKE in the Mediterranean Sea over 1993-2023, from all-sat European product. White arrows represent the circulation of Atlantic waters, adapted from Millot and Taupier-Letage (2005). Boxes represent the regions analyzed in Fig. 7, the Alboran Sea in red and Ierapetra area in blue.

## 2 Data and methods

### 2.1 Altimetry data derived products

#### 2.1.1 Level 4

We rely on three gridded Level 4 (L4) satellite altimetry products provided by the E.U. Copernicus Marine Service (CMEMS: https://marine.copernicus.eu/, Le Traon et al., 2019). These products include gap-free daily zonal and meridional geostrophic velocity anomalies derived from gridded Sea Level Anomaly (SLA) fields:

- "**all-sat-glo**": a global 1/4° resolution product based on all available altimeter missions, ranging from two in 1993 to up to seven in 2023 (Fig. 2);

- "**all-sat-euro**": using all available altimeter missions as all-sat-glo, but interpolated onto a 1/8° resolution grid of the European region;

- "**two-sat-glo**": a global 1/4° resolution product derived from a two-satellite constellation. This configuration ensures temporal stability and homogeneity, making it particularly suitable for climate studies.

#### 2.1.2 Level 3

Level 3 (L3) along-track data is also used for comparison with the L4 products. Also available from CMEMS, we specifically select tracks corresponding to the reference altimetry missions (TOPEX, Jason series, Sentinel-6). This selection ensures spatial



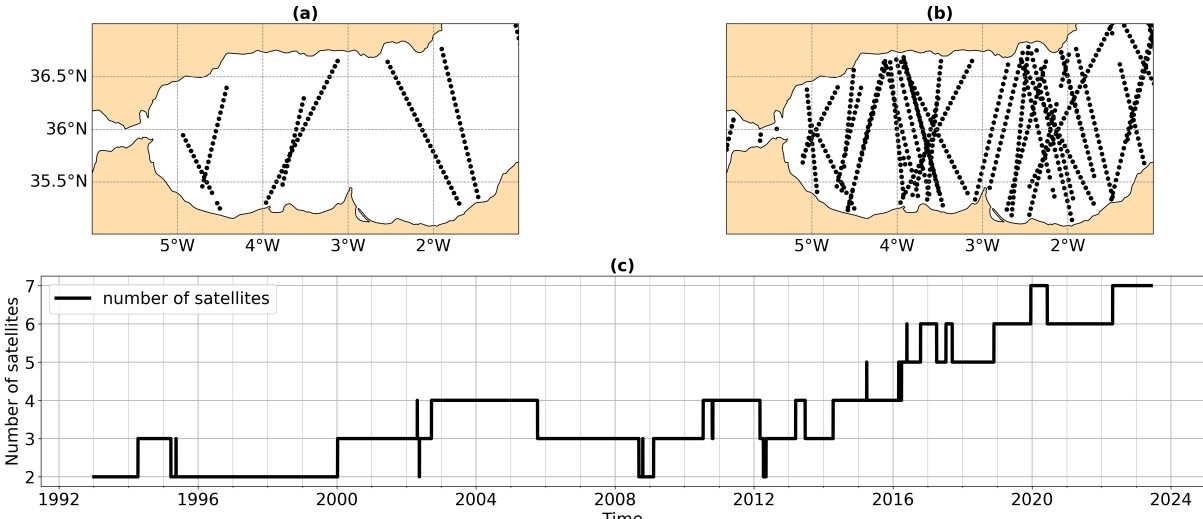

**Figure 2. (a) Ground tracks from the two satellites used in two-sat-glo over 20 days (from May $1^{st}$, 2022 to May $20^{th}$, 2022) in the Alboran Sea. (b) Same as in (a) for all-sat products (7 satellites between May $1^{st}$, 2022 and May $20^{th}$, 2022). (c) Evolution of the number of altimetry missions used to build the all-sat products.**

and temporal consistency of the tracks throughout the 1993–2023 period. This data set provides along-track SLA measurements with a temporal resolution of 10 days, and approximately 7 km for the along-track spatial resolution. In the remainder of this manuscript, this dataset will be referred to as **L3-ref**.

### 2.1.3 Eddy Atlas

We use the all-sat and two-sat altimetric Mesoscale Eddy Trajectories Atlas (META3.2 DT), produced by SSALTO/DU-ACS and distributed by AVISO+ (https://aviso.altimetry.fr). These atlas identify all anticyclonic and cyclonic eddies detected from the gridded all-sat-glo and two-sat-glo data sets, and provide key eddy characteristics including location, closed contours, radius, and rotational speed. The detection algorithm used to generate this atlas was developed in collaboration between IMEDEA (Mason et al., 2014) and CLS, is freely available under GNU General Public License https://github.com/AntSimi/py-eddy-tracker.

### 2.2 Methods

#### 2.2.1 Eddy Kinetic Energy

Throughout the manuscript the EKE is defined as:

$$EKE = \frac{1}{2}\left(u_{ga}^2 + v_{ga}^2\right) \tag{1}$$



where $u_{ga}$ and $v_{ga}$ refer to the components of geostrophic velocity anomalies in the zonal and meridional directions, respectively. Note that in the original CMEMS products, geostrophic velocity anomalies ($u_{ga}$, $v_{ga}$) are computed relative to the reference period 1993–2012. For consistency with the full duration of the altimetric record, we recalculated these anomalies using an extended reference period of 1993–2023, so that they now reflect deviations from the mean over the entire altimetry era.

### 2.2.2 Along-track orthogonal velocities

For the L3-ref product, we use the component of the geostrophic velocity orthogonal to the altimeter track ($\underline{u_{ga\perp}}$) derived from the SLA along-track data ($\eta$). $\underline{u_{ga\perp}}$ is computed using a central finite difference method with a 5-point stencil width as follow (in the horizontal plane):

$$\underline{u_{ga\perp,i}} = \frac{g}{f} \cdot \frac{-\eta_{i+2} + 8\eta_{i+1} - 8\eta_{i-1} + \eta_{i-2}}{12l} \cdot \begin{pmatrix} -\sin(\alpha) \\ \cos(\alpha) \end{pmatrix} \tag{2}$$

Where $l$ is the distance between two successive nadir measurements, $\alpha$ is the angle between the north direction and the satellite track, $g$ is the gravitational acceleration, and $f$ is the Coriolis parameter. The EKE computed from L3 data ($EKE_\perp = \frac{1}{2}u_{ga\perp}^2$) is thus based solely on the orthogonal component of the geostrophic velocity anomalies and therefore represents a portion of the total EKE. For comparison, we also calculated $EKE_\perp$ from the L4 data by interpolating the gridded SLA field onto the along-track positions (both in space and time).

### 2.2.3 Mann-Kendall test

All calculated trends presented in this study are assessed using the Mann-Kendall test (MK test). MK test is a non-parametric method commonly used to detect monotonic trends in time series. However, serial autocorrelation in the data affects the significant level of the MK test. To address this, we adopted the correction proposed by Yue and Wang (2004) implemented in the Pymannkendall Python package (https://pypi.org/project/pymannkendall/). Moreover, the standards errors ($SE$) associated with the different trend estimates were evaluated as the residual standard error, considering the effective sample size (Stan Development Team, 2021), and normalized by the temporal spread of the data points (James et al., 2023):

$$SE = \frac{\sqrt{\frac{\sum(y_i - \hat{y}_i)^2}{n_{eff} - 2}}}{\sqrt{\sum(x_i - \bar{x})^2}} \tag{3}$$

With $y_i$ the observations, $\hat{y}_i$ the predictions, $n_{eff}$ the effective sample size, $x_i$ the independent variable (i.e. time), and $\bar{x}$ its mean. Trends were considered statistically significant when the associated p-value was below $0.05$.



## 3 Results

### 3.1 Mean Eddy Kinetic Energy in the Mediterranean Sea

The mean EKE in the Mediterranean Sea exhibits a heterogeneous spatial distribution, with most of the basin characterized by
relatively low values of EKE (Fig. 1). The overall spatiotemporal mean is $66.10\ cm^2s^{-2}$. In the western Mediterranean, the
Alboran Sea stands out with intense eddy activity around the two semi-permanent anticyclonic gyres (Renault et al., 2012),
reaching a spatiotemporal mean EKE of $235.58\ cm^2s^{-2}$. This vigorous eddy activity, up to $400\ cm^2s^{-2}$ (Mason et al., 2023),
plays a key role in modulating heat and salt transport through the Strait of Gibraltar (Bryden et al., 1994; Tsimplis and Bryden,
2000; Sánchez-Román et al., 2009). In the eastern Mediterranean, the most energetic feature is the Ierapetra eddy, a long-lived
anticyclonic structure that forms southeast of Crete (Larnicol et al., 2002; Ioannou et al., 2017; Pegliasco et al., 2021), with a
spatiotemporal mean EKE of $319.63\ cm^2s^{-2}$, up to $500\ cm^2s^{-2}$ (Mason et al., 2023). For comparison, EKE values over the
Gulf Stream and the Kuroshio Extension can exceed $3000\ cm^2s^{-2}$ (Renault et al., 2017; Barceló-Llull et al., 2025).

### 3.2 Influence of altimeter coverage: all-sat versus two-sat

The EKE averaged over the Mediterranean Sea (Fig. 3a) from all-sat-glo shows a significant increase during the altimetric
era, from a mean of $44.91\ cm^2s^{-2}$ in 1993 to $68.04\ cm^2s^{-2}$ in 2022. The computed trend is $0.75\pm0.06\ cm^2s^{-2}year^{-1}$
(Fig. 3d) and is statistically significant. This result is consistent with all-sat-euro which exhibits a similar trend of $0.87\pm$
$0.05\ cm^2s^{-2}year^{-1}$ (Fig. 3d). Although comparable, the all-sat-euro trend is slightly stronger as it has a higher spatial reso-
lution (1/8° compared to 1/4° for all-sat-glo), which allows for better representation of mesoscale eddies and their associated
energy (Wang et al., 2022). However, the two-sat-glo product shows a markedly weaker and not statistically significant trend
of $0.06\pm0.04\ cm^2s^{-2}year^{-1}$ over the Mediterranean Sea (Fig. 3a and 3d). The larger trend detected in the all-sat products
can be explained by the fact that they estimate nearly twice as much EKE as the two-sat-glo product in recent years. Despite
these notable differences in trends magnitude, all three L4 products show strong agreement on seasonal cycles and interannual
variability ($r = 0.98$ between all-sat-glo and all-sat-euro and $r = 0.79$ between all-sat-glo and two-sat-glo after removing their
linear trends).
In the Alboran Sea region (Fig. 3b and 3d), all three products exhibit positive and statistically significant EKE trends, in
contrast to the Mediterranean basin as a whole where the two-sat-glo trend is not significant. The trends are also higher in
magnitude than those observed for the whole Mediterranean, with a particularly pronounced gap between all-sat-glo ($3.04\pm$
$0.23\ cm^2s^{-2}year^{-1}$) and two-sat-glo ($0.43\pm0.18\ cm^2s^{-2}year^{-1}$). This indicates that the impact of the increasing number
of satellites is especially evident in energetic regions such as the Alboran, where all-sat products capture much stronger EKE
levels than the two-sat product. A focus on the eastern Alboran gyre reveals similar trends to those observed across the broader
Alboran Sea for all-sat products ($3.64\pm0.32\ cm^2s^{-2}year^{-1}$ for all-sat-euro and $3.44\pm0.31\ cm^2s^{-2}year^{-1}$ for all-sat-glo),
while revealing a net increase for the two-sat-glo product: $1.16\pm0.28\ cm^2s^{-2}year^{-1}$. This suggests that the alignment of
altimeter tracks relative to energetic features (Fig. 2) plays a significant role in capturing variability accurately.
Unlike the other regions, the Ierapetra region displays strong statistically significant negative trends (Fig. 3c and 3d). In addi-



tion, the EKE time series reveal intense peaks compared to the rest of the basin. The Ierapetra eddy itself is not a permanent feature (Ioannou et al., 2017), but rather a seasonal anticyclonic structure with peak intensity in late summer (Fig. 3c). We notice a marked decrease in peak intensity, with EKE maxima reaching $\sim 1500\ cm^2 s^{-2}$ before 2007 but only $\sim 500\ cm^2 s^{-2}$ after 2012. In this region, all products capture similar trends, likely because the spatial coverage of the two-sat product is already good near a crossover point of the reference altimeter tracks, which enhances the accuracy of the gridded fields and

reduces interpolation errors (Pascual et al., 2007). Additionally, the lower short-term variability in this region (compared to the Alboran Sea) means that the temporal resolution of the two-sat product is sufficient to capture the SLA signal accurately, thereby narrowing the difference with the all-sat products.







**Figure 3.** Area-weighted mean EKE over (a) the Mediterranean Sea, (b) the Alboran Sea and (c) the Ierapetra area for the different L4 products. Thinner lines are the raw daily data while thicker lines represent the yearly-rolling mean. Straight lines correspond to the trends of the raw data. The time series are calculated by averaging the EKE over the regions indicated in the insets, where the black lines are the reference altimeter tracks. (d) EKE trends and their associated errors for the different L4 products in the three studied regions. Statistically non-significant trends are indicated in italics (only two-sat-glo in the Mediterranean Sea).



Part of the differences observed between EKE trends computed with all-sat and two-sat altimetric products over the Mediter-
ranean Sea could be explained by the increasing number of merged altimeters in the all-sat products, which may amplify EKE
signals over time. To further explore this hypothesis, we examined the temporal evolution of EKE differences between products
in relation to the number of satellites incorporated in all-sat. The evolution of the difference in the area-weighted mean EKE
over the Mediterranean Sea between all-sat-glo and two-sat-glo exhibits a marked increase over time (Fig. 4a). This increase is
strongly correlated with the temporal evolution of the number of altimetry missions merged into the all-sat-glo product, with a
Pearson correlation coefficient of $0.85$. As more altimeters are included into the all-sat product, spatial and temporal sampling
improve, which enhance the detection of mesoscale signals (Ballarotta et al., 2019). This improved observational capability
introduces an artificial positive trend of EKE. This strong correlation suggests that a significant portion of the observed trend
in all-sat products seems to arise from the increasing number of satellites, rather than from changes in oceanic variability. Note
that once more than five altimeters are operational, the rate of increase in the EKE difference slows down. To isolate this artifi-
cial trend, we performed a second degree polynomial regression between the signal difference and the number of satellites. By
subtracting this fitted signal from the original all-sat-glo time series, we obtained a version of the product without the artificial
trend, shown in green in Fig. 4b. This new signal, denoted as all-sat-glo-corrected, represents the all-sat-glo EKE with the
artificial satellite-driven trend removed. The resulting trend for all-sat-glo-corrected is $0.12\ cm^2s^{-2}year^{-1}$, and statistically
significant, which is slightly higher than the trend of two-sat-glo of $0.06\ cm^2s^{-2}year^{-1}$.

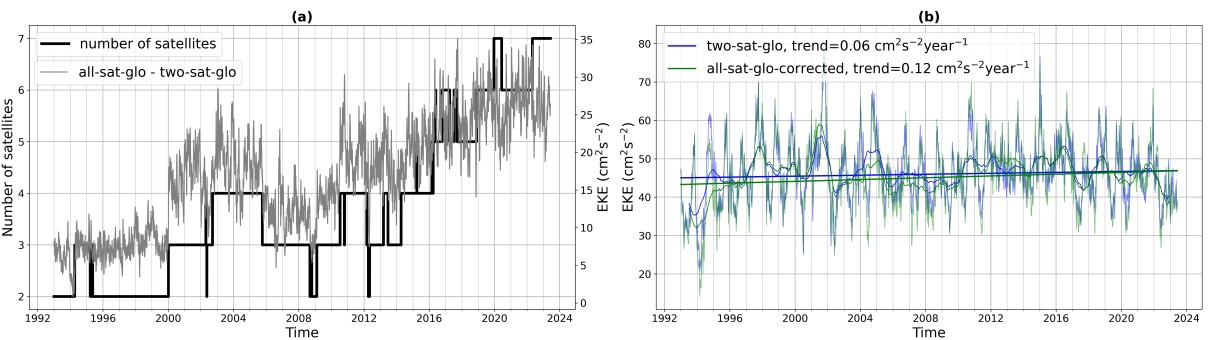

**Figure 4. (a) Difference between all-sat-glo and two-sat-glo EKE along the number of altimetry missions. (b) Time series of the
area-weighted mean EKE over the Mediterranean Sea for two-sat-glo and all-sat-glo-corrected.**

To identify the regions that are more likely to be affected by the impact of an increasing number of satellites, we computed
the Pearson correlation coefficient between the all-sat-glo and two-sat-glo differences and the number of altimetry missions
at each grid point (Fig. 5). High correlations are especially pronounced in low-energy areas (Fig. 1) such as the Adriatic
Sea, Tyrrhenian Sea, eastern Levantine basin, and along the Libyan coast. These areas coincide with zones where all-sat-glo
trends are statistically significant while two-sat-glo trends are not (Fig. 6). This spatial coherence highlights that, in these low-
energy regions, the trends observed in the all-sat products are most likely driven by the increasing number of merged satellite
altimeters. In such regions, the signal-to-noise ratio is lower, making it more sensitive to observational changes. Moreover, the
Alboran Sea, and in particular the western Alboran gyre, also exhibits high correlation values. This is likely explained not only





by the limited sampling of the gyre structure by the two-sat tracks, but also by the highly dynamic nature of the region as the entry point of the Atlantic Jet (Renault et al., 2012). The complex and rapidly evolving circulation is not fully captured by the two-sat configuration. In such case, the additional spatial and temporal coverage provided by multiple satellites substantially

improves the reconstruction of the dynamics, thereby increasing the observed correlation.

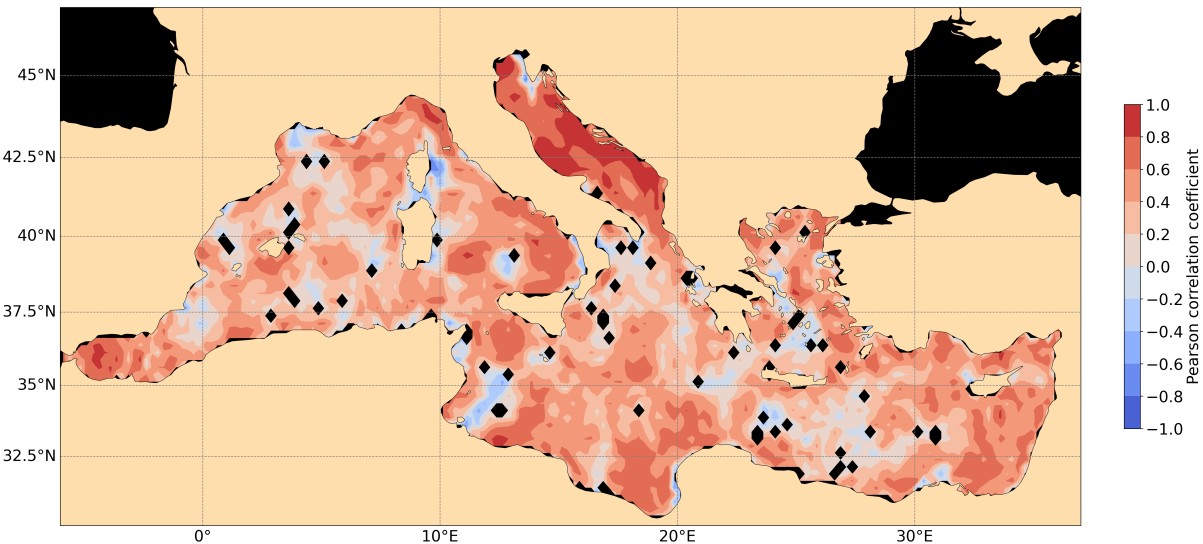

**Figure 5.** Spatial correlation map between all-sat-glo and two-sat-glo EKE differences and the number of altimetry missions. Black dots indicate non-significant correlations. A one-month low-pass filter was applied to the time series at each grid point.

These spatial correlations between EKE differences and satellite coverage offer insight into where artificial amplification of EKE trends is most likely to occur. To place these findings in a broader spatial context, we now examine the full two-dimensional distribution of EKE trends across the Mediterranean Sea. These maps (Fig. 6) reveal that most of the Mediterranean Sea exhibits positive EKE trends along the main southern surface currents, especially pronounced in regions such as the Alboran

Sea, along the north African coast, and in the vicinity of the Mersa-Matruh area (southeast of the Ierapetra eddy). In contrast, the Ierapetra eddy is the only region showing strong statistically significant negative EKE trends. Even though, the northern Ionian Sea also reveals negative trends while the central Ionian Sea shows positive trends, possibly due to a change in the Ionian Sea circulation from anticyclonic to cyclonic (Bessières et al., 2013; Kalimeris and Kassis, 2020). All products present a similar pattern with stronger trends for all-sat products. What differs substantially is the spatial coverage of statistically

significant trends between the all-sat and two-sat products (Fig. 6). Although $72.40\%$ and $71.27\%$ of the Mediterranean have significant trends for all-sat-euro and all-sat-glo, respectively, only $48.70\%$ is significant for two-sat-glo.



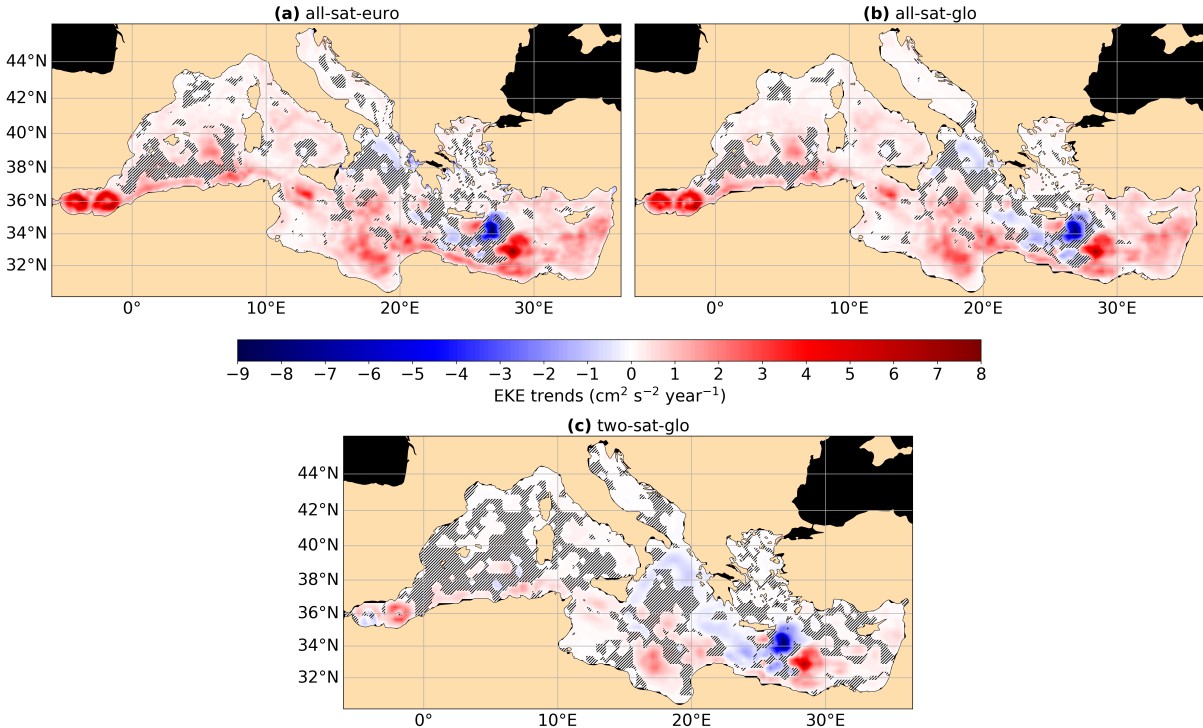

**Figure 6.** EKE trends in the Mediterranean Sea for (a) all-sat-euro, (b) all-sat-glo and (c) two-sat-glo. Hatched areas are statistically non-significant trends.

**Table 1.** Different trends depending on the data set and the method. EKE in the first raw: the anomalies were computed using 1993-2023 as the reference period. EKE in the second raw: the EKE was computed using raw geostrophic velocity anomalies from the CMEMS data sets (1993-2012 reference period for the SLA).

| Trends in $cm^2\,s^{-2}\,year^{-1}$ | | all-sat-euro | all-sat-glo | two-sat-glo |
|---|---|---|---|---|
| **EKE** **(1993-2023 anomalies)** | Mean of all trends | 0.43 | 0.38 | 0.06 |
| | Mean of the significant trends | 0.59 | 0.52 | 0.12 |
| | Trend of the mean | 0.87 | 0.75 | 0.06 |
| **EKE** **(1993-2012 CMEMS reference period)** | Mean of all trends | 0.52 | 0.46 | 0.11 |
| | Mean of the significant trends | 0.67 | 0.60 | 0.21 |
| | Trend of the mean | 1.03 | 0.90 | 0.17 |

To understand how these regional differences influence basin-scale assessments of EKE variability, we compare trends as the spatial average of local trends with trends derived from the area-weighted mean EKE time series (Table 1). This comparison highlights the influence of high-energy regions on basin-wide trends. In all-sat products, the trend of the area-weighted mean

EKE time series is higher than the mean of the trends because high-energy regions, though spatially limited, dominate the




average time series due to their strong EKE values and pronounced trends. In contrast, this behavior is not observed for two-sat because of weaker trends in energetic regions. These energetic areas are better resolved in all-sat, leading to greater influence on the overall trend. We also distinguish these trends in two categories with different reference periods to compute the anomalies: the entire period (1993-2023) that are the results presented in this manuscript (first raw in Table 1) and 1993-2012 which is the reference period used in the CMEMS products (second raw in Table 1). We systematically observe higher trends for the shorter reference period, suggesting an increase in kinetic energy in recent years and highlighting the need to compute geostrophic anomalies with respect to the full time period when computing eddy kinetic energy.

## 3.3 Eddy Kinetic Energy trends in high-energy regions

To evaluate the fidelity of the gridded products in capturing mesoscale variability, we compared the L3 and L4 data sets in two high-energy regions of the Mediterranean Sea (Fig. 7), the Alboran Sea and the Ierapetra area, assessing how well the gridded fields reproduce the observed variability.

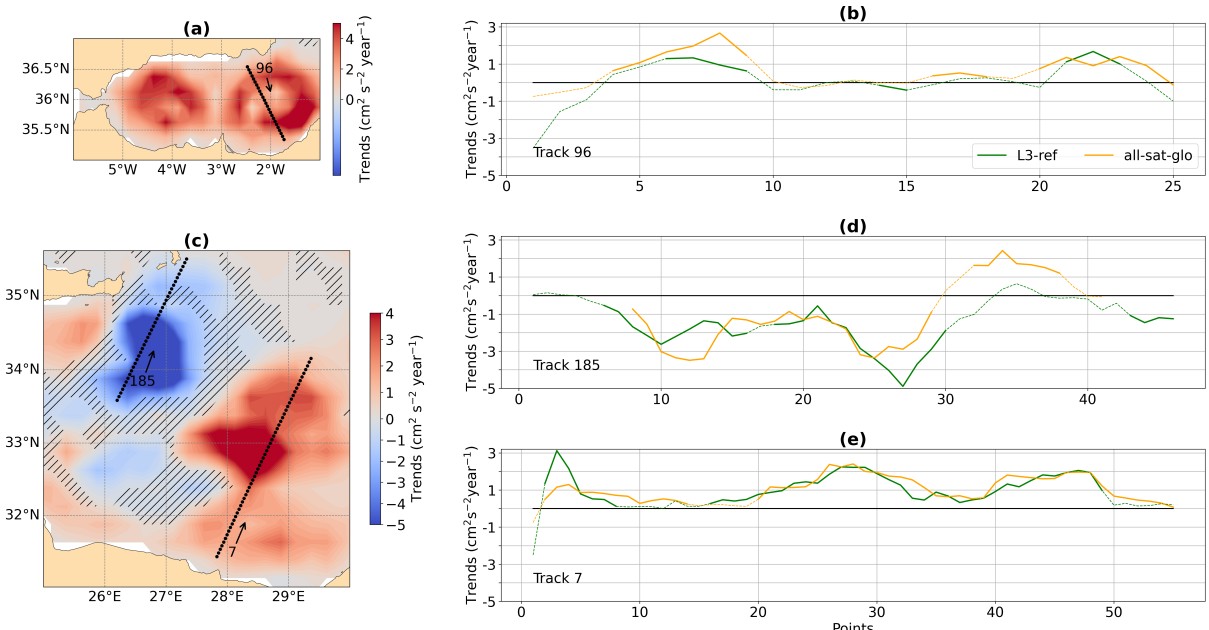

**Figure 7. (a) EKE trends in Alboran Sea (1993-2023), red box in Fig. 1. Background colors represent the gridded EKE trends from all-sat-glo. Hatched areas highlight regions where trends are not statistically significant. The overlaid dots depict reference altimeters tracks with arrows indicating the direction of the satellite tracks and the number indicating their name. (b) Along-track $EKE_\perp$ trends in Alboran Sea for track 96. Dotted lines correspond to non-significant trends while solid lines to the significant trends. (c) Same as in (a) but for the Ierapetra region (Ierapetra eddy and Mersa-Matruh), blue box in Fig. 1. (d) and (e) Same as in (b) but for the Ierapetra region, respectively track 185 and track 7.**



In addition to exhibiting high EKE values, the Alboran Sea region displays strong, positive, and statistically significant trends in EKE over the studied period (Fig. 7a and 7b). Notably, one of the reference satellite altimetry mission tracks nearly crosses the center of the eastern Alboran gyre (track 96; Fig. 7a). In this configuration, the geostrophic velocity component
along the track is close to zero, while the perpendicular component dominates the total geostrophic velocity. Therefore, the perpendicular EKE along this track ($EKE_\perp$) is expected to be very close to the total EKE, and thus enhances the reliability of the observed trend. In fact, we note the similarity of the trends of $EKE_\perp$ between the L3-ref and all-sat-glo products for this track (Fig. 7b). This result indicates that the processing used to produce gridded L4 products does not introduce significant errors in EKE estimates along-tracks, supporting the suitability of L4 data for studying mesoscale variability in regions with
dense track coverage. A closer examination reveals that the strongest positive and significant trends are located at the edges of the gyre rather than at its core (Fig. 7a and 7b), where horizontal velocities and their variability are typically greater (Pujol and Larnicol, 2005; Barceló-Llull et al., 2017), leading here to enhanced positive $EKE_\perp$ trends.

In contrast, the Ierapetra eddy area, crossed by track 185, shows a negative trend in EKE (Fig. 7c and 7d). As observed in the Alboran region, the $EKE_\perp$ trends are consistent between the L3 and L4 products. Southeast of the Ierapetra eddy lies the
Mersa-Matruh area, characterized by episodic rather than persistent eddy activity (Pujol and Larnicol, 2005; Barboni et al., 2021) and reveals an increase in mesoscale activity (track 7, Fig. 7c and 7e). The agreement in $EKE_\perp$ trends across products further confirms the robustness of the results.



## 3.4 Comparison of eddy characteristics

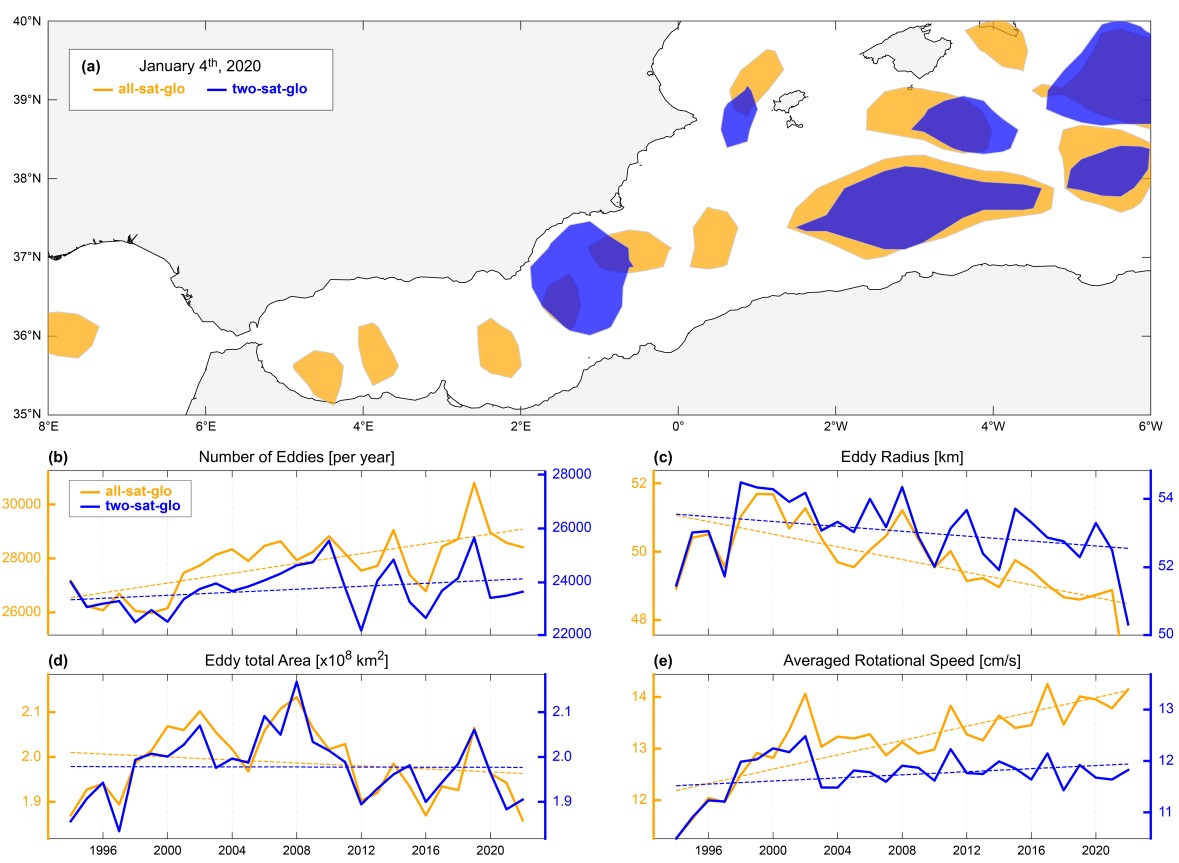

**Figure 8. (a) Eddies detected on January $4^{th}$, 2020 using all-sat-glo (blue) and two-sat-glo (yellow). (b) Annual number of detected eddies. (c) Annual mean eddy radius. (d) Total annual eddy-covered area. (e) Annual mean eddy speed.**

EKE captures the overall intensity of oceanic mesoscale variability, integrating contributions from a variety of dynamical
processes, including frontal instabilities, current meandering, and coherent mesoscale eddies. Among these, ocean coherent
eddies are of particularly interest as they are long-lived, rotating water masses that trap and transport heat, salt, and nutrients
across vast distances (Barceló-Llull et al., 2017; Barabinot et al., 2024).

In this section, we focus on the statistics of these mesoscale eddies derived from the all-sat-glo and two-sat-glo satellite
products, including their number, size, spatial extent, and rotational speed (Fig. 8). A representative snapshot of detected
eddies from each dataset on January 4, 2020, illustrates clear differences with the all-sat-glo product capturing a larger number
of eddies and finer-scale structures compared to two-sat-glo (Fig. 8a). Over the altimetric era, the number of eddies detected
per year shows a stronger increasing trend in all-sat-glo ($27812$ per year on average) than in to two-sat-glo ($23744$ per year,
Fig. 8b). In parallel, the decreasing trend in the eddy radius size in the all-sat-glo product (Fig. 8c) reflects its improved ability





to detect smaller-scale features. Nonetheless, the total area occupied by eddies remains comparable between the two data
sets, with no significant trend and similar variability detected in both products (Fig. 8d), indicating a balance between the
higher number of eddies and their reduced size. Finally, the average eddy rotational velocity (Fig. 8e), highly relevant to local
biogeochemical variability, shows a significant increasing trend in the all-sat-glo data, while no such trend is evident in two-sat.
In general, these results indicate that the progressive increase in the number of altimetric satellites has enhanced the detection
of mesoscale activity (Amores et al., 2019), leading to more numerous, smaller, and faster eddies in the all-sat-glo product,
while the total eddy-covered area remains unchanged. These discrepancies also mirrors the EKE results and further highlights
the limitations of the two-satellite configuration in resolving the diversity of mesoscale processes in the Mediterranean Sea,
consistent with earlier work by Pascual et al. (2007).

## 4   Discussion

### 4.1   Regional variability and changes in EKE

The Alboran gyres and the Mersa-Matruh area confirm a general intensification of EKE in high-energy areas, consistent with
patterns observed in global high-energy regions (Martínez-Moreno et al., 2021; Barceló-Llull et al., 2025). However, trends
over the Mediterranean Sea are not spatially uniform and the Ierapetra area is a high-energy region showing negative EKE
trends.

The literature describing Alboran gyres is abundant (Heburn and La Violette, 1990; Renault et al., 2012; Sánchez-Garrido
et al., 2013; Juza et al., 2016; Sánchez-Garrido and Nadal, 2022). Although transient events have been observed (Viúdez et al.,
1998; Vélez-Belchí et al., 2005; Peliz et al., 2013), they are mainly persistent gyres with strong seasonal variability (stronger in
summer) and are one of the most important signals observed in the Mediterranean Sea. The strong positive trends highlighted
at their edges (Fig. 6 and 7a) reflect their interannual modulation (Iudicone et al., 1998; Vargas-Yáñez et al., 2002; Pujol and
Larnicol, 2005; Sánchez-Garrido and Nadal, 2022). As seen in Fig. 3b the variability in EKE increases during the altimetric
record (standard deviation of $44.24\ cm^2s^{-2}$ for the period 1993-2008 and $59.04\ cm^2s^{-2}$ for the period 2013-2023 for the
Alboran Sea). There is still a gap in understanding the drivers of this variability in the circulation of the Alboran Sea, which
include variations in the flow through the Strait of Gibraltar, local winds, and background stratification (Peliz et al., 2013;
Sánchez-Garrido and Nadal, 2022). For example, previous studies have indicated that the eastern gyre is less stable and may
disappear or shift under specific conditions, such as during winter and spring when its kinetic energy decreases to nearly zero
and it is replaced by the central cyclonic gyre under moderate Atlantic inflow, or through interactions with the western gyre
(Vargas-Yáñez et al., 2002; Peliz et al., 2013). In terms of EKE, the most energetic scenario in the area is the eastward migra-
tion of the western Alboran gyre before its fusion with the eastern Alboran gyre (Peliz et al., 2013). Peliz et al. (2013) noted
that these migration events were associated with instabilities of the Atlantic Jet, more precisely south shifts of the Atlantic Jet
along the Moroccan coasts, which appear to be related to the establishment of easterlies (Bolado-Penagos et al., 2021).

In the Levantine basin, the Ierapetra eddy is the only clear seasonal signal (Larnicol et al., 2002; Menna et al., 2012), although
it does not appear every year (Mkhinini et al., 2014; Ioannou et al., 2017). Studies have highlighted the role of the interaction



of Etesian winds (persistent northerly summer winds over the Aegean Sea) with Crete's orography in its formation and inten-
sification (Horton et al., 1997; Amitai et al., 2010; Ioannou et al., 2020). The Ierapetra eddy region is associated with a strong
negative EKE trends (Fig. 6). Figure 3c shows a net decrease also in the intensity of the EKE peaks. In fact, Gandham et al.
(2025) have observed a decline in Etesian wind episodes across the eastern Mediterranean and we surmise that this decrease in
wind episodes could damp the development of the Ierapetra eddy.

Concerning the Mersa-Matruh area, this region acts as an attractor (Menna et al., 2012; Barboni et al., 2021) of transient eddies
without clear boundaries, but characterized by a relatively high EKE activity (Fig. 1 and Pujol and Larnicol, 2005). Ierape-
tra eddies tend to go west along the Libyo-Egyptian current (Sutyrin et al., 2009). However, some go south-east towards the
Mersa-Matruh area (Ioannou et al., 2017) in what seemed to be an isolated behavior. The loss of intensity of the Ierapetra
eddies could modify their trajectories and enhance the activity in the Mersa-Matruh area.

## 4.2   Influence of altimetry products and methods

A central finding of this study is the discrepancy in the magnitude of the EKE trend between altimetry products based on
different satellite constellations (Fig. 3). In high-energy regions we showed that the intensification in the Alboran Sea and
the decline in the Ierapetra area are supported by robust and statistically significant trends (Fig. 3 and 6). However, when
extending the analysis to the entire Mediterranean basin, the trend obtained with two-sat-glo is statistically not significant
($0.06 \pm 0.04$ $cm^2.s^{-2}.year^{-1}$). While the all-sat products exhibit positive trends over the whole basin, the inconsistency
between products and the limited significance of two-sat-glo trend make it difficult to confidently assert a basin-scale increase in
EKE. Moreover, the EKE values derived from all-sat products show a strong correlation with the increasing number of merged
satellite altimeters (Fig. 4 and 5). This increase in merged satellites improves spatial and temporal resolution (Ballarotta et al.,
2019) with time, introducing an artificial trend, especially in low-energy areas. Eddy characteristics shown in Fig. 8 have also
highlighted this artificial trend: the number of detected eddies increases with time, whereas the total area they occupy remains
approximately constant. This suggests that the enhanced temporal and spatial sampling of all-sat products, arising from the
growing number of satellites, enables the detection of smaller eddies that were previously unresolved. Nonetheless, part of the
differences in EKE trends can also be attributed to the two-sat product being derived from a two-satellite constellation and, as
demonstrated by Pascual et al. (2007), a minimum of three concurrent altimeter missions are required to adequately monitor
mesoscale variability in the Mediterranean Sea.

## 5   Conclusions

We have analyzed three decades of satellite altimetry data (1993-2023) to assess whether surface ocean dynamics in the
Mediterranean Sea is intensifying over time, focusing on the temporal evolution of Eddy Kinetic Energy (EKE). We have
computed EKE trends from three gridded altimetric products: a global product derived from a stable two-satellite constellation
(two-sat) and two others (global and European) incorporating all available satellites (all-sat). The robustness of the trends
computed from the gridded products has been tested by comparing them with original along-track measurements in high-energy



areas. These high-energy areas revealed significant positive (Alboran Sea, Mersa-Matruh) and negative (Ierapetra area) EKE
trends for all products. However, the increasing number of satellite altimeter missions merged into all-sat products influences
the magnitude of EKE trends, as does the relative to tracks position of the structures studied. In fact, the correlation found
between the all-sat-glo/two-sat-glo EKE difference and the evolution of the number of satellite altimetry missions merged in
all-sat-glo suggests an artificial trend in all-sat products caused by this increasing number of missions incorporated. Moreover,
in regions of low-energy, the trends are not significant in two-sat product but are significant in all-sat suggesting that, in these
regions, trends are mainly artificial and due to the merging of new satellites in all-sat products. It is important when studying
long-term ocean variability analysis to use data set built for climate applications as the two-sat product. Finally, the study of
eddy characteristics raises the difficulty of two-sat to capture the mesoscale in the Mediterranean Sea. Pascual et al. (2007)
concluded that at least three altimeters are necessary to accurately monitor mesoscale activity in the Mediterranean Sea due to
the smaller typical size of the structures. Building on this, we suggest that a three-sat product available since 2000 would be a
valuable tool to further assess EKE variability in the Mediterranean Sea.

*Data availability.* The altimetric data products used in this study are publicly available on the E.U. Copernicus Marine Service Information
website (CMEMS: https://marine.copernicus.eu/, Le Traon et al., 2019). The gridded products were downloaded in July 2024, their version
is "vDT2021", and is described in their respective QUality Information Document (QUID) (Pujol et al., 2023). The vDT2021 all-sat-euro
product is available with product ID: SEALEVEL_EUR_PHY_L4_MY_008_068, DOI: https://doi.org/10.48670/moi-00141. The vDT2021
all-sat-glo product is available with product ID: SEALEVEL_GLO_PHY_L4_MY_008_047, DOI: https://doi.org/10.48670/moi-00148. The
vDT2021 two-sat-glo product is available with product ID: SEALEVEL_GLO_PHY_CLIMATE_L4_MY_008_057, DOI: https://doi.org/10.
48670/moi-00145. Along-track data are also available via the CMEMS website with product ID: SEALEVEL_EUR_PHY_L3_MY_008_061
and DOI: https://doi.org/10.48670/moi-00139. The altimetric Mesoscale Eddy Trajectories Atlas (META3.2 DT), is produced by SSALTO/-
DUACS and distributed by AVISO+ (https://aviso.altimetry.fr) with support from CNES, in collaboration with IMEDEA (DOI: https:
//doi.org/10.24400/527896/a01-2022.005.220209 for the META3.2 DT all-sat-glo version and https://doi.org/10.24400/527896/a01-2022.
006.220209 for the META3.2 DT two-sat-glo version).

*Author contributions.* All of the authors conceptualized the study. P.H. performed the data analysis and wrote the first draft. All of the authors
contributed to the preparation of the final draft.

*Competing interests.* The contact author has declared that none of the authors has any competing interests.



*Disclaimer.* The author(s) employed an artificial intelligence–assisted language editing tool to enhance the clarity and readability of the manuscript. All content was subsequently reviewed, revised, and approved by the author(s), who assume full responsibility for the final version of the work.

*Acknowledgements.* The present research was carried out within the framework of the activities of the Spanish Government through the "María de Maeztu Centre of Excellence" accreditation to IMEDEA (CSIC-UIB) (CEX2021-001198). ObsSea4Clim "Ocean observations and 330 indicators for climate and assessments" is funded by the European Union, Horizon Europe Funding Programme for Research and Innovation under grant agreement number: 101136548. ObsSea4Clim contribution nr. 25. P.H. is funded by ObsSea4Clim. V.C. acknowledges the support from the Spanish Ramón y Cajal Program (RYC2020-029306-I) through Grant AEI/UIB—10.13039/501100011033. B. B.-L. is funded by the Balearic Government Vicenç Mut program (Grant num. PD/008/2022) and acknowledges support from the METARAOR Project (Grant num. PID2022-139349OB-I00) funded by MCIN/AEI/10.13039/501100011033/FEDER, UE.



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
