# Peer review of "Eddy Kinetic Energy Variability From 30 Years of Altimetry in the Mediterranean Sea"

_EGUsphere, 2025_

## Referee Comment (RC3)

**General Comment**

The manuscript assesses trend in EKE in the Mediterranean Sea over the 1993-2023 using gridded altimetry products (a two-satellite product and two all-satellite products). The authors examine both basin-wide and regional EKE changes, analyse differences between products, investigate eddy characteristics, validate trends against along-track data and explore the influence of sampling changes associated with the increasing number of altimeters.

The manuscript is clearly written and the results are clearly presented supported by thoughtful figures.

This work follows closely the recent global study by Barceló-Lull et al. (2025). The present manuscript applies a very similar methodological framework - contrasting two satellite and all-satellite altimetric products and analysing regional EKE trends — but within the specific context of the Mediterranean Sea. In doing so, it provides a valuable regional application that complement the global perspective.

I recommend its publication with minor revisions.

*Barceló-Llull, B., Rosselló, P., Combes, V., Sánchez-Román, A., Pujol, M. I., and Pascual, A.: Kuroshio Extension and Gulf Stream dominate the Eddy Kinetic Energy intensification observed in the global ocean, Sci. Rep-UK, 15, https://doi.org/10.1038/s41598-025-06149-9,2025.*

**Specific Comments**

**Line 92**: For clarity and consistency with common altimetry literature, the authors may consider referring to this component as the cross-track velocity.

**Lines 140-143**: The study reports trends for the eastern Alboran gyre but no figure specifically highlights this sub-region. Including a small inset or zoomed panel for the eastern gyre would help readers better visualise these local trends.

**Figure 3**: Please defined "area-weighted mean". A brief explanation of how the weighting is performed would improve clarity.

**Figure 4**: I suggest slightly rephrasing the caption for panel (a) for clarity. For example: "Time series of the difference between all-sat-glo and two-sat-glo EKE and the number of altimetry missions."

**Lines 186-188**: "*Even though, the norther nIonian Sea … from anticyclonic to cyclonic.*"
The sentence is not very clear. I suggest the following rephrasing:

"The northern Ionian Sea also shows negative trends, whereas the central Ionian displays positive trends, a pattern that may be linked to shifts in the basin's circulation between anticyclonic and cyclonic states (Bessières et al., 2013; Kalimeris and Kassis, 2020)."

**Table 1**: The terms "mean of all trends" and "trend of the mean" should be defined.

**Line 193**: As mentioned in a previous comment, the term "area-weighted mean" has not been defined and should be clarified.

**Line 200**: The motivation for recomputing EKE using the CMEMS 1993–2012 reference period is not fully explained. It is a bit unclear to me what additional insight it provides beyond showing the sensitivity of trends to anomaly definition. The need to compute geostrophic anomalies with respect to the full time period is standard practice.

**Figure 7(b)**: We observe a negative trend in the L3-ref date that does not appear in the L4 product. Could the authors provide an explanation for this discrepancy?

**Line 211**: I would suggest using the term "cross-track" instead of "perpendicular".

**Section 3.4**: The manuscript reports statistics for "mesoscale eddies" in general, without distinguishing cyclonic vs anticyclonic. EKE contributions and dynamical behaviour can differ between cyclones and anticyclones. The authors could provide separate statistics if possible. Even a brief note would strengthen the interpretation.

---

## Author Comment (AC1)

We would like to thank the reviewer for their constructive and insightful comments on our manuscript. We are grateful for the time and expertise invested in evaluating our work.

Their suggestions have helped us clarify and strengthen several aspects of the manuscript, particularly regarding the use of META eddy statistics, the interpretation of regional case studies, and the broader relevance of our findings.

Below, we provide a detailed point-by-point response to each of the reviewer's comments, along with a summary of the corresponding modifications implemented in the revised manuscript.

Eddy Kinetic Energy (EKE) is a widely used metric for ocean currents variability, enabling to monitor them using remote-sensing, particularly through altimetry since 1993. Some earlier studies reported increasing trends in EKE worldwide, which was not yet extensively studied in the Mediterranean Sea. The authors explored the sensibility to the increasing number of altimetric satellites. Section 3.2 explores the differences between all-sat and two-sat Level 4 products ; section 3.3 asserts the results by comparison with along-tracks Level 3 data ; section 4 provides an insight of the statistics using the META eddy atlas.

The method provides an insight to altimetry products biases, often curtailed in many mesoscale studies. The results are quite interesting as they show the number of satellites drives a strong bias on observed long-term trends, in EKE but also potentially on individual eddy detections. They showed that while an EKE trend might appear over the Mediterranean Sea in all-sat product, this is actually not significant using two-sat timeseries. Figure 5 in particular is very interesting - and might be further highlighted - as a mapped sensitivity of satellite sampling.

Section 3,4 using META eddy detections is interesting as it asserts the previous results with eddies statistics. It seems however to stay under-used, as it introduces a totally different dataset for 15 lines of development. I recommend extending this section to illustrates other regions and stregnthen the robustness of the statistics, and also make distinction between cyclones and anticyclone (not possible using EKE only).

A major concern arises about the regions of application, as the study focuses on 2 limited energetic regions to assess its results. These areas (Alboran & Ierapetra) are known to be energetic because of the presence of a single (or two) recurrent mesoscale structures. In the particular case of Ierapetra the reported results seem to be a shift of the structure.

**Major comment :**

- Adding an application of the method to larger areas where mesoscale structures are more numerous and behavior more chaotic should be more significant. Such areas in the Mediterranean Sea could be for instance the Algerian Basin or the Central Ionian Jet. Also increase the area of the Ierapetra focus, for instance by a Levantine Basin-wide focus.

We focused on Alboran and Ierapetra because they are the regions with the highest EKE and there the trends are relatively important. Moreover, those two areas presented significant trends in both all-sat and two-sat products. Finally, the presence of recurrent mesoscale structures allows us to monitor changes in their behavior.

An analysis centered on the Ionian Sea, the eastern Algerian Basin and the Levantine Sea is now provided in Supplementary materials (Fig. S1, included below).

Considering the Ionian Sea (Fig. S1a), trends are positive for all-sat products and negative for two-sat and significant for all of them. In the northern Ionian Sea (Fig. S1b) trends are statistically non-significant for all-sat products and significant and negative for two-sat (Fig. S1e). This suggests that trends for all-sat products in the Ionian Sea are dominated by positive trends in the southern Ionian Sea (confirmed by Fig. 6a and Fig.6b). During the year 1997, there was a change in the Ionian Sea circulation, from anticyclonic to cyclonic (Bessières et al., 2012), relocating the relatively high-energy area of the Ionian Sea from the north to the south. This created a drop of EKE visible in Fig. S1b. We also added a discussion about this in the last paragraph of section 4.1 (L. 305-317).

Considering a larger area for the entire Levantine basin (Fig. S1c) and studying the eastern Algerian basin (area with positive trends in Fig. 6; Fig. S1d), the results of the trends of the different products (Fig. S1e) show the same pattern as for the entire Mediterranean Sea or the Alboran Sea, i.e. high positive trends for all-sat products while insignificant or low for two-sat.

Bessières, L., M.H. Rio, C. Dufau, C. Boone, M.I. Pujol, 2012: Ocean state indicators from MyOcean altimeter products, Ocean Sciences, doi:10.5194/osd-9-2081-2012

[Figure]

Figure S1: Area-weighted mean EKE over (a) the Ionian Sea, (b) the northern Ionian Sea, (c) the Levantine Sea and (d) the eastern Algerian basin for the different L4 products. Thinner lines are the raw daily data while thicker lines represent the yearly-rolling mean. Straight lines correspond to the trends of the raw data. The time series are calculated by averaging the EKE over the regions indicated in the insets, where the black lines are the reference altimeter tracks. (e) EKE trends and their associated errors for the different L4 products in the four studied regions. Statistically non-significant trends are indicated in italics and bars are gray-hatched.

**Additional comments :**

- l.29 'A widely used metric […] is the EKE' : Here it would be helpful to make reference to previous well-known studies to better introduce EKE. For example, Wilkin & Morrow (JGR,1994), Stammer (JPO,1997) or Ferrari & Wunsch (2009).

We agree and added Wilkin and Morrow, 1994 (L. 30).

- l.40-50 (but also l.309) : when introducing the Mediterranean Sea, remind that the Rossby deformation radius is much smaller in this area, ranging between 20 and 10km. This area is not well described in Chelton et al (1998), but one can also refer to Kurkin et al (https://doi.org/10.2205/2020ES000737, 2020)

We added the following sentence (L. 43-45): **The Mediterranean is also distinguished by a first baroclinic Rossby deformation radius ranging from about 10 to 20 km (Beuvier et al., 2012; Kurkin et al., 2020), indicating that mesoscale features are smaller and challenging to observe from satellite data.**

- Fig.1 : add isolines of EKE to better reflect gradients

We added isolines of EKE to Fig. 1.

- l.140 : 'A focus on the Eastern Alboran gyre' : not shown

We added the trends for the eastern Alboran gyre as a subplot of Fig. 3 (Fig. 3c).

- l.147 : For the Ierapetra region : Can you add a description of your region boundary ? It sounds quite small, and therefore may be quite sensitive to shift in boundaries. A more objective box might be drawn taking an EKE-isoline as boundary

We agree and changed the definition of the boundary of the Ierapetra area taking an EKE-isoline and a brief description was added to the text (L. 154-156):

Unlike the other regions, the Ierapetra region displays strong statistically significant negative trends (Fig. 3d and 3e, **the boundary of the Ierapetra area was defined as an EKE-isoline of the 30 years mean of all-sat-euro product in order to have the same region for all the products, value of the isoline at 200 cm²s⁻².**

- l.156 (and after Table.1, l.193) : The 'area-weighted mean' was never defined

We added a definition of "area-weighted mean" in section 2.2.1 (L. 99-101).

- l.164 : can you add this fitted polynomial to the plot ? If Correlation coeff is so high (0,85), why not using a linear fit instead, as number of satellite is already roughly linear ?

We agree that a linear regression could have been used. We performed the analysis using both linear and polynomial regressions and obtained similar results. The figure below compares the two fitting methods.

[Figure]

- l.171 : The case of the Alboran sea (high EKE but high sensibility) can be a counter-example. Isn't it rather almost land-locked areas that are more sensitive ? (Alboran, Adriatic, Levantine basin?)

We agree that being land-locked is one of the factors contributing to a region's sensitivity to the increasing number of satellites. In addition, the orientation of the tracks relative to the surface currents—specifically to compute the cross-track velocities—also (maybe more?) plays an important role. We have modified the text accordingly (L. 189-198):

**Moreover, the Alboran Sea, although a high-energy area, also presents a high correlation. High correlation values are found mainly in the western Alboran gyre, while the eastern gyre exhibits lower correlations. This contrast is likely linked to the orientation of the two-satellite tracks relative to the surface currents. In the eastern gyre, one of the two-sat tracks crosses close to the gyre center, providing better sampling of the geostrophic velocities. In the western gyre, however, the tracks mostly sample the edges of the gyre, leading to limited representation of its surface geostrophic velocities. This uneven sampling is further exacerbated by the highly dynamic nature of the western Alboran gyre, located at the entry point of the Atlantic Jet (Renault et al., 2012), where the circulation evolves rapidly and is not fully captured by the two-sat configuration. Consequently, the additional spatial and temporal coverage provided by multiple satellites substantially improves the reconstruction of the dynamics in this region, thereby increasing the observed correlation.**

- l.188 : about reversal of Ionian Sea circulation, reference about BIOS : Gacic et al (2010)

We added the reference (L. 206).

- Table1 : Not clear what are defined as 'Mean of all trends' and 'Trend of the mean'.

We added the following description (L. 212-214): **'Mean of all trends' corresponds to the spatial average of all grid-point trends. 'Trend of the mean' represents the linear trend computed from the area-weighted EKE time series averaged over the whole Mediterranean basin, the one shown in Fig. 3.**

- l.234 : 'ability to detect smaller scales features' This can be asserted by references such as Amores et al (2018) or Stegner et al (2021).

We added the paper from Amores et al., 2018 (L. 257).

- Section3.4 : as described earlier, this subsection is quite interesting but deserves to be investigated further and applied to other regions. A distinction between cyclone and anticyclone should also be made, at least in figure 8. Actually, it sounds surprising that region with cyclonic activity (e.g. South-West Cretean gyre, Rhodes gyre) do not exhibit more sensitivity to increasing number of satellite (in your Fig.5), as they tend be smaller (see observing system experiment in Stegner et al 2021). Do you think EKE trends are rather driven by anticyclone increasing rotating speed over time, as it might be the case for the Alboran gyres ?

We thank the reviewer for this comment. We separated cyclones from anticyclones in our analysis, modified Fig. 8, and discussed the results in section 3.4 (L. 249-269):

**In this section, we focus on the statistics of these mesoscale eddies derived from the all-sat-glo and two-sat-glo satellite products, distinguishing cyclonic from anticyclonic eddies, including their number, size, spatial extent, and rotational speed (Fig. 8). A representative snapshot of detected eddies from each dataset on January 4, 2020, illustrates clear differences with the all-sat-glo product capturing a larger number of eddies and finer-scale structures compared to two-sat-glo (Fig. 8a).**

**Over the altimetric era, the number of eddies detected per year shows a stronger increasing trend in all-sat-glo (27812 per year on average) than in two-sat-glo (23744 per year, Fig. 8b). For both products, a higher number of cyclones than anticyclones is detected with 57% of the eddies being cyclonic, in agreement with Pegliasco et al., 2021. In parallel, the decreasing trend in the eddy radius size in the all-sat-glo product for both cyclonic and anticyclonic structures (Fig. 8c) reflects its improved ability to detect smaller-scale features (Amores et al., 2018). In contrast, in the two-sat-glo product only anticyclones show a decrease in radius, while cyclones do not display any detectable trend (Fig. 8c).**

**Despite these negative trends for the mean radius, when eddies are not separated by polarity, the total eddy area remains approximately constant and similar between the two products. However, separating cyclones from anticyclones reveals opposite tendencies in the two-sat-glo product (Fig. 8d): the total area covered by anticyclones decreases, while the area associated with cyclones increases. No marked polarity-dependent trends are observed in the all-sat product.**

**Finally, the average eddy rotational velocity (Fig. 8e), highly relevant to local biogeochemical variability, shows significant increasing trends in the all-sat-glo data, with a stronger increase for anticyclones, while no such trend is evident in two-sat.**

**In general, these results indicate that the progressive increase in the number of altimetric satellites has enhanced the detection of mesoscale activity (Amores et al., 2018), leading to more numerous, smaller, and faster eddies in the all-sat-glo product, while the total eddy-covered area remains unchanged. These discrepancies also**

mirror the EKE results and further highlights the limitations of the two-satellite configuration in resolving the diversity of mesoscale processes in the Mediterranean Sea, consistent with earlier work by Pascual et al., 2007.

We further expanded our discussion in section 4.2 (L. 328-354):

Eddy characteristics shown in Fig. 8 have also highlighted this artificial trend: the number of detected eddies increases with time for all-sat-glo, whereas the total area they occupy remains approximately constant. This suggests that the enhanced temporal and spatial sampling of all-sat products, arising from the growing number of satellites, enables the detection of smaller eddies that were previously unresolved.

Moreover, the distinction between cyclonic and anticyclonic eddies reveals several polarity-dependent differences in the long-term evolution of eddy statistics (Fig. 8), pointing out intrinsic differences in their detectability and dynamical behavior. Overall, more cyclones are detected (57 % Fig. 8 and Pegliasco et al., 2021). This asymmetry is fully consistent with the known differences in the reliability of eddy detection in gridded altimetric products. Stegner et al., 2021 have shown that anticyclones in the Mediterranean are detected with high positional accuracy and moderate radius biases, while cyclones, particularly large ones, are substantially less reliable, with larger position errors, stronger overestimation of radius (only category without negative trend in Fig. 8c), and greater sensitivity to interpolation artifacts. These detection biases arise from fundamental dynamical differences: large anticyclones tend to be more coherent and longer-lived, whereas cyclones are more unstable and prone to splitting into smaller, fast-evolving sub-mesoscale structures that are poorly resolved by altimetric gridding.

The two-sat product, which maintains a stable configuration over time, further supports this interpretation. While it shows no significant trends in eddy number or speed, separating polarities reveals a positive trend in cyclones total eddy area. This pattern is consistent with the "coarsening artifact" described in Stegner et al., 2021, where small structures are smoothed into larger, spurious cyclonic features during interpolation.

These polarity-dependent behaviors have direct implications for the interpretation of EKE trends in gridded altimetric products. The stronger increase in mean rotational speed for anticyclones than for cyclones in all-sat suggests that the positive EKE trends reported in multi-mission products are driven by anticyclonic intensification, as observed in the semi-permanent Alboran gyres.

As a result, long-term EKE trends in all-sat configurations should be interpreted with caution, as they may reflect improved sampling of anticyclones rather than energetic changes in the mesoscale field. Nonetheless, part of the differences in EKE trends can also be attributed to the two-sat product being derived from a two-satellite constellation and, as demonstrated by Pascual et al., 2007, a minimum of three concurrent altimeter missions are required to adequately monitor mesoscale variability in the Mediterranean Sea.

- Section 4.4 : you can add some discussion also about decennal cycles in the Ionian Sea, where reversal of the regional circulation are expected to have some effects on the EKE of the Midde Mediterranean Jet.

We added the following paragraph to the discussion in section 4.1 (L. 305-317):

**In addition to the regional patterns discussed above, the Ionian Sea is known to exhibit pronounced decadal oscillations in its surface and intermediate circulation, commonly referred to as the BiOS (Bimodal Oscillating System; Gacic et al., 2010, Bessieres et al., 2013). These recurrent transitions between cyclonic and anticyclonic circulation regimes can substantially modulate the intensity and position of the Mid-Mediterranean Jet (MMJ). During cyclonic phases, the MMJ tends to strengthen and shift southward, enhancing mesoscale activity along its path, whereas anticyclonic phases are associated with a weakening and northward shift of the jet (Bessieres et al., 2013). Such regime shifts can therefore imprint decadal variability on EKE in the Ionian Sea. This is consistent with the spatially heterogeneous trend patterns apparent in Fig. 6. In all three products, the Ionian region displays strong spatial contrasts, with negative EKE trends south of the Peloponnese and west of Greece, while positive trends emerge in the central and southern Ionian. The gray-hatched regions, indicating non-significant trends, are also more prevalent in the central Ionian, reinforcing the idea that this area is subject to decadal variability rather than a monotonic long-term change (Fig. S1 in the Supplement). Taken together, these patterns are compatible with a transition from anticyclonic to cyclonic circulation in the Ionian Sea. In particular, Gačić et al. (2010) reported a marked regime shift in 1997, when the circulation flipped from anticyclonic to cyclonic.**

- l.275-276 : This behavior could be checked using the META atlas

We include a video (video_S1.mp4) and a figure (Fig. S2), which is a snapshot of the video, in Supplementary materials to check this behavior. We thank the reviewer for suggesting the use of the META atlas. We have performed a dedicated analysis of eddy trajectories using the META atlas.

We identified long-lived anticyclonic eddies in the Ierapetra region using the META 3.2 delayed-time (DT) atlas built from the all-sat-glo product. To specifically target Ierapetra eddies, we defined a geographical selection box (red dashed rectangle) corresponding to the region where these eddies are known to form. We then selected anticyclonic eddies whose first detection occurs within this box and whose lifetime exceeds 10 days. For each selected anticyclone, the black line represents the full trajectory from its first to its last detection. The colored circles correspond to the individual eddy detections during the last 10 days of the eddy lifetime, with the color indicating the eddy age in days. This representation highlights both the propagation pathways of long-lived Ierapetra anticyclones and their final evolution prior to dissipation.

This analysis does not support a systematic southeastward propagation of Ierapetra eddies toward the Mersa-Matruh area these last years. The manuscript has been revised accordingly, and the results now indicate that Ierapetra eddies have become shorter-lived and smaller in recent years.

We hope these revisions adequately address the reviewer's concerns and improve the clarity and scientific contribution of our manuscript. We remain grateful for the helpful feedback.

Sincerely,

Paul Hargous, on behalf of all co-authors

---

## Author Comment (AC2)

We thank the reviewer for their positive evaluation of our manuscript. We are grateful that the reviewer supports publication of the manuscript after minor revisions.

Below, we provide a detailed point-by-point response to each of the reviewer's comments, along with a summary of the corresponding modifications implemented in the revised manuscript.

Satellite altimetry products from 1993 to 2023 are assessed to quantify the temporal evolution of surface geostrophic current variability in the Mediterranean Sea. In particular, trends in Eddy Kinetic Energy (EKE) are estimated using three gridded altimetric products. Along-track datasets and an eddy atlas are also employed for comparison in selected regions. Significant positive and negative EKE trends are identified in the most energetic areas of the basin. The influence of the increasing number of satellites on the observed EKE rise is evaluated. Products based on a constant two-satellite configuration appear adequate for estimating long-term trends, but they fail to capture important dynamical structures in some regions. Therefore, the use of constant three-satellite altimetry products is recommended for future investigations of EKE variability in the Mediterranean Sea.

The paper is well structured, and the English is generally fluent and clear. The figures effectively support the written statements. I recommend the publication of this manuscript in Ocean Science, after the authors address the following specific comments (minor revision).

Specific comments:

L43 and L45. Change "waters" to "water".

We modified it (L. 46, 48).

L45. Remove "and freshwater".

Removed (L. 48).

L47. Add the following reference before Escudier et al. : Poulain, P. M., Menna, M., & Mauri, E. (2012). Surface geostrophic circulation of the Mediterranean Sea derived from drifter and satellite altimeter data. *Journal of Physical Oceanography*, *42*(6), 973-990. Add this reference in the reference list.

We added the reference (L. 49).

L76. Provide mode details on the Eddy Atlas. Is it based on ADT or SLA structures?

We added a description of the atlases in section 2.1.3 (L. 85-89) and the reference of Pegliasco et al., 2022 (L. 80):

**The META eddy atlases are constructed from gridded Absolute Dynamic Topography (ADT) fields (i.e., SLA + MDT, Mean Dynamic Topography) from DUACS. A high-pass filter removes large-scale signals to isolate mesoscale variability and eddies are then identified from closed ADT contours using strict geometric and amplitude criteria. Finally, eddies are tracked in time by overlapping their effective contours between consecutive days to build trajectories.**

Caption of Fig. 3., L156 and L193. Change "Area-weighted mean" to "Area-averaged".

We acknowledge the potential confusion between "area-weighted mean" and "area-averaged." The definition of the area-weighted mean has now been added to the Data and Methods section in 2.2.1 (L. 99-101). Note that because the region of interest is relatively small, the area of individual grid points varies only slightly across the Mediterranean Sea; as a result, the area-weighted mean and the simple area-averaged are nearly identical.

In this analysis we computed the area-weighted mean (L. 99-101) as it is scientifically more correct (also used in previous studies: Barceló-Llull et al., 2025; Martínez-Moreno et al., 2021).

Caption of Fig. 6. Change to "Gray-hatched".

We corrected it.

Table 1. Define "Trend of the mean".

We added the following description (L. 212-214): **"Mean of all trends" corresponds to the spatial average of all grid-point trends. "Trend of the mean" represents the linear trend computed from the basin-wide area-weighted EKE time series, the one shown in Fig. 3.**

Caption of Fig. 7. Add info about the L3-ref and all-sat-glo curves.

Added. **Green corresponds to L3-ref trends and yellow to all-sat-glo ones.**

L216. Change "horizontal velocities and their variability are" to "geostrophic velocity variability is".

Corrected (L. 237).

L274. Change "However, some go south-east" to "However, some move south-east".

Corrected (L. 303).

L306. Change "applications as the two-sat product" to "applications such as the two-sat product".

Changed (L. 368).

We hope these revisions adequately address the reviewer's concerns and improve the clarity and scientific contribution of our manuscript. We remain grateful for the helpful feedback.

Sincerely,

Paul Hargous, on behalf of all co-authors

---

## Author Comment (AC3)

We thank the reviewer for their positive assessment of our manuscript. We appreciate the reviewer's acknowledgement of the connection with the recent global study by Barceló-Llull et al. (2025) and their appreciation of the added value provided by our regional focus on the Mediterranean Sea.

Below, we provide a detailed point-by-point response to each of the reviewer's comments, along with a summary of the corresponding modifications implemented in the revised manuscript.

General Comment

The manuscript assesses trend in EKE in the Mediterranean Sea over the 1993-2023 using gridded altimetry products (a two-satellite product and two all-satellite products). The authors examine both basin-wide and regional EKE changes, analyse differences between products, investigate eddy characteristics, validate trends against along-track data and explore the influence of sampling changes associated with the increasing number of altimeters.

The manuscript is clearly written and the results are clearly presented supported by thoughtful figures.

This work follows closely the recent global study by Barceló-Lull et al. (2025). The present manuscript applies a very similar methodological framework - contrasting two satellite and all-satellite altimetric products and analysing regional EKE trends — but within the specific context of the Mediterranean Sea. In doing so, it provides a valuable regional application that complement the global perspective.

I recommend its publication with minor revisions.

Barceló-Llull, B., Rosselló, P., Combes, V., Sánchez-Román, A., Pujol, M. I., and Pascual, A.: Kuroshio Extension and Gulf Stream dominate the Eddy Kinetic Energy intensification observed in the global ocean, Sci. Rep-UK, 15, https://doi.org/10.1038/s41598-025-06149-9,2025.

Specific Comments

Line 92: For clarity and consistency with common altimetry literature, the authors may consider referring to this component as the cross-track velocity.
We agreed and changed "orthogonal" for "cross-track".

Lines 140-143: The study reports trends for the eastern Alboran gyre but no figure specifically highlights this sub-region. Including a small inset or zoomed panel for the eastern gyre would help readers better visualise these local trends.
We added the trends for the eastern Alboran gyre as a subplot of Fig. 3 (Fig. 3c).

Figure 3: Please defined "area-weighted mean". A brief explanation of how the weighting is performed would improve clarity.
We added a definition of "area-weighted mean" in section 2.2.1.

Figure 4: I suggest slightly rephrasing the caption for panel (a) for clarity. For example: "Time series of the difference between all-sat-glo and two-sat-glo EKE and the number of altimetry missions."

We agree and corrected it.

Lines 186-188: "Even though, the norther nIonian Sea ... from anticyclonic to cyclonic." The sentence is not very clear. I suggest the following rephrasing: "The northern Ionian Sea also shows negative trends, whereas the central Ionian displays positive trends, a pattern that may be linked to shifts in the basin's circulation between anticyclonic and cyclonic states (Bessières et al., 2013; Kalimeris and Kassis, 2020)."

We included this change as it makes the text clearer (L. 204-207).

Table 1: The terms "mean of all trends" and "trend of the mean" should be defined.

We have clarified these definitions.

The manuscript now includes the following text (L. 212-214): **"Mean of all trends" corresponds to the spatial average of all grid-point trends. "Trend of the mean" represents the linear trend computed from the basin-wide area-weighted EKE time series, the one shown in Fig. 3.**

Line 193: As mentioned in a previous comment, the term "area-weighted mean" has not been defined and should be clarified.

Added in section 2.2.1.

Line 200: The motivation for recomputing EKE using the CMEMS 1993–2012 reference period is not fully explained. It is a bit unclear to me what additional insight it provides beyond showing the sensitivity of trends to anomaly definition. The need to compute geostrophic anomalies with respect to the full time period is standard practice.

We recomputed anomalies over the period 1993–2023 to ensure that the temporal mean of the anomaly is zero. We added the analyses on the CMEMS provided data (reference period 1993-2012) to evaluate a possible change in the magnitude of the trends. We have noticed that a large number of studies compute EKE using the anomalies directly provided by CMEMS, which are based on a non-centered reference period (1993–2012) (L. 95-98).

Figure 7(b): We observe a negative trend in the L3-ref date that does not appear in the L4 product. Could the authors provide an explanation for this discrepancy?

If the reviewer refers to the beginning of track 96 in Fig. 7b (first five points), the discrepancy may be related to the proximity of the satellite measurements to the coast, a region where altimetric data generally have increased uncertainties.

If the reviewer refers to the middle of the track (around point 15), the L3 product represents single-track, pointwise measurements without spatial averaging. As a result, the EKE at this location is highly sensitive to small shifts in the gyre position or structure. In contrast, the L4 fields involve spatial interpolation and smoothing, which tend to attenuate very localized variability. This difference in the product could explain the small difference in the trends obtained there.

Line 211: I would suggest using the term "cross-track" instead of "perpendicular".

We modified it.

Section 3.4: The manuscript reports statistics for "mesoscale eddies" in general, without distinguishing cyclonic vs anticyclonic. EKE contributions and dynamical behaviour can differ between cyclones and anticyclones. The authors could provide separate statistics if possible. Even a brief note would strengthen the interpretation.

We agree with the reviewer and separated cyclones from anticyclones in our analysis and modified Fig. 8 and the results description in section 3.4 (L. 249-269):

In this section, we focus on the statistics of these mesoscale eddies derived from the all-sat-glo and two-sat-glo satellite products, distinguishing cyclonic from anticyclonic eddies, including their number, size, spatial extent, and rotational speed (Fig. 8). A representative snapshot of detected eddies from each dataset on January 4, 2020, illustrates clear differences with the all-sat-glo product capturing a larger number of eddies and finer-scale structures compared to two-sat-glo (Fig. 8a).

Over the altimetric era, the number of eddies detected per year shows a stronger increasing trend in all-sat-glo (27812 per year on average) than in two-sat-glo (23744 per year, Fig. 8b). For both products, a higher number of cyclones than anticyclones is detected with 57% of the eddies being cyclonic, in agreement with Pegliasco et al., 2021. In parallel, the decreasing trend in the eddy radius size in the all-sat-glo product for both cyclonic and anticyclonic structures (Fig. 8c) reflects its improved ability to detect smaller-scale features (Amores et al., 2018). In contrast, in the two-sat-glo product only anticyclones show a decrease in radius, while cyclones do not display any detectable trend (Fig. 8c).

Despite these negative trends for the mean radius, when eddies are not separated by polarity, the total eddy area remains approximately constant and similar between the two products. However, separating cyclones from anticyclones reveals opposite tendencies in the two-sat-glo product (Fig. 8d): the total area covered by anticyclones decreases, while the area associated with cyclones increases. No marked polarity-dependent trends are observed in the all-sat product.

Finally, the average eddy rotational velocity (Fig. 8e), highly relevant to local biogeochemical variability, shows significant increasing trends in the all-sat-glo data, with a stronger increase for anticyclones, while no such trend is evident in two-sat.

In general, these results indicate that the progressive increase in the number of altimetric satellites has enhanced the detection of mesoscale activity (Amores et al., 2018), leading to more numerous, smaller, and faster eddies in the all-sat-glo product, while the total eddy-covered area remains unchanged. These discrepancies also mirror the EKE results and further highlights the limitations of the two-satellite configuration in resolving the diversity of mesoscale processes in the Mediterranean Sea, consistent with earlier work by Pascual et al., 2007.

We also deepen our discussion in section 4.2 (L. 328-354):

Eddy characteristics shown in Fig. 8 have also highlighted this artificial trend: the number of detected eddies increases with time for all-sat-glo, whereas the total area they occupy remains approximately constant. This suggests that the enhanced

temporal and spatial sampling of all-sat products, arising from the growing number of satellites, enables the detection of smaller eddies that were previously unresolved.

Moreover, the distinction between cyclonic and anticyclonic eddies reveals several polarity-dependent differences in the long-term evolution of eddy statistics (Fig. 8), pointing out intrinsic differences in their detectability and dynamical behavior. Overall, more cyclones are detected (57 % Fig. 8 and Pegliasco et al., 2021). This asymmetry is fully consistent with the known differences in the reliability of eddy detection in gridded altimetric products.

In fact, Stegner et al., 2021 have shown that anticyclones in the Mediterranean are detected with high positional accuracy and moderate radius biases, while cyclones, particularly large ones, are substantially less reliable, with larger position errors, stronger overestimation of radius (only category without negative trend in Fig. 8c), and greater sensitivity to interpolation artifacts. These detection biases arise from fundamental dynamical differences: large anticyclones tend to be more coherent and longer-lived, whereas cyclones are more unstable and prone to splitting into smaller, fast-evolving sub-mesoscale structures that are poorly resolved by altimetric gridding.

The two-sat product, which maintains a stable configuration over time, further supports this interpretation. While it shows no significant trends in eddy number or speed, separating polarities reveals positive trend in cyclones total eddy area. This pattern is consistent with the "coarsening artifact" described in Stegner et al., 2021, where small structures are smoothed into larger, spurious cyclonic features during interpolation.

These polarity-dependent behaviors have direct implications for the interpretation of EKE trends in gridded altimetric products. The stronger increase in mean rotational speed for anticyclones than for cyclones in all-sat suggests that the positive EKE trends reported in multi-mission products are driven by anticyclonic intensification, as observed in the semi-permanent Alboran gyres.

As a result, long-term EKE trends in all-sat configurations should be interpreted with caution, as they may reflect improved sampling of anticyclones rather than energetic changes in the mesoscale field. Nonetheless, part of the differences in EKE trends can also be attributed to the two-sat product being derived from a two-satellite constellation and, as demonstrated by Pascual et al., 2007, a minimum of three concurrent altimeter missions are required to adequately monitor mesoscale variability in the Mediterranean Sea.

We hope these revisions adequately address the reviewer's concerns and improve the clarity and scientific contribution of our manuscript. We remain grateful for the helpful feedback.

Sincerely,

Paul Hargous, on behalf of all co-authors

---

## Author Response (AR2)

We thank the reviewer for their positive evaluation of our manuscript. We are grateful that the reviewer supports publication of the manuscript after minor revisions.

Below, we provide a detailed point-by-point response to each of the reviewer's comments, along with a summary of the corresponding modifications implemented in the revised manuscript.

The authors answered the referree comments in an adequate way. Of particular interest is the extended analysis in Sect 3.4 about mesoscale eddies distinguishing anticyclones and cyclones. This further highlights the differences between all-sat and two-sat products, but using the framework of coherent mesoscale eddies.

The paragraphs added in the discussion also further explain the observed difference between two-sat and all-sat products. The processes can be either regional pattern changes (Sect4.1) or altimetry processing artefacts (Sect4.2)

Below are some minor specific comments.

l.89 : Is there any filter on eddy lifetime to be considered as a track ?
In the META eddy atlas, eddies are sorted in three categories, including short-lived eddies (lifetime strictly shorter than 10 days), long-lived eddies (longer than 10 days), and untracked eddies. Untracked eddies correspond to features detected at a given time step that are not associated with any other eddy at adjacent time steps. All categories of detected eddies are included in our analysis. We added the following sentence (l. 89-91):
**In this study, no lifetime threshold is applied and all detected eddies are considered, including short-lived eddies (lifetime < 10 days), long-lived eddies (> 10 days), and untracked eddies, which are detected features not associated with any other eddy in time.**

l.249 "in this section" : remind for the reader that you will use here the META Atlas, as the data source varies between Sect 3.3 and 3.4.
We modified the text as follow (l. 251-253):
**In this section, we use the META Atlas to analyze the statistics of these mesoscale eddies derived from the all-sat-glo and two-sat-glo satellite products, distinguishing cyclonic from anticyclonic eddies, including their number, size, spatial extent, and rotational speed (Fig. 8).**

l.253-254 : number of eddies per year is still given adding AE and CE, confusing with Fig.8b
We now specify in the text (l. 256-257):
**Over the altimetric era, the total number of eddies detected per year shows a stronger increasing trend in all-sat-glo (27812 per year on average) than in two-sat-glo (23744 per year). Figure 8b further differentiates between anticyclonic and cyclonic eddies.**

l.247 : Discussing specifically observation of watermasses trapped by mesoscale eddies, one can also refer to Barboni et al (2023) in the Mediterranean Sea, or Laxenaire et al (2019) for Agulhas Rings.
Thank you. We added Barboni et al. 2023.

l.303 : "Previous studies have suggested that some
Ierapetra eddies may deviate south-eastward toward the Mersa-Matruh area" you can add that you checked this in a Lagrangian Framework in your supplementary material. It shows that this behavior is quite isolated, hence the EKE trend dipole (Fig.7c) is not a shift of the same mesoscale structure. This seems a fair result to add.

We completed the text (l. 308-310):

**We investigated this behavior using the META atlas (see Supplementary Material), which shows that such deviations are relatively isolated events. This indicates that the EKE trend dipole observed in the area (Fig. 7c) does not reflect a systematic shift of the same mesoscale structure.**

l.373-375 : Nice addition to your conclusion. You can also add (or maybe at the end of discussion) that the same is valid not only for EKE but also to intrepete mesoscale eddies long-term trends. The use of META two-sat product is then advised.

As altimetric products evolve, differences between climate-oriented and multi-mission data sets require caution when interpreting long-term trends in EKE **and mesoscale eddy parameters**. While the Surface Water and Ocean Topography mission (SWOT; Morrow et al., 2019; Fu et al., 2024) will enhance mesoscale observations (Wang et al., 2025; Verger-Miralles et al., 2025), sustained altimetric continuity remains essential to understand the drivers and impacts of ocean energetics in the Mediterranean Sea and the global ocean.

We hope these revisions adequately address the reviewer's concerns and improve the clarity and scientific contribution of our manuscript. We remain grateful for the helpful feedback.

Sincerely,

Paul Hargous, on behalf of all co-authors